# The Impact of Household Chaos and Dietary Intake on Executive Function in Young Children

**DOI:** 10.3390/nu13124442

**Published:** 2021-12-12

**Authors:** Samantha Iwinski, Sharon M. Donovan, Barbara Fiese, Kelly Bost

**Affiliations:** 1Department of Human Development and Family Studies, University of Illinois, Urbana, IL 61801, USA; bhfiese@illinois.edu (B.F.); kbost@illinois.edu (K.B.); 2Division of Nutritional Sciences, University of Illinois, Urbana, IL 61801, USA; sdonovan@illinois.edu; 3Department of Food Science and Human Nutrition, University of Illinois, Urbana, IL 61801, USA

**Keywords:** dietary intake, executive function, household chaos, young children, parenting, STRONG Kids2

## Abstract

Children’s executive functions (EFs) emerge over time and can be shaped by household environments and dietary intake. However, there is a lack of knowledge about how these factors influence EFs in children aged 18–24 months. This study tested a model exploring the relations between parent-reported dietary intake, household chaos, and child EF. The sample consisted of 294 families participating in the STRONG Kids2 birth cohort study of nutrition and child health. Caregivers completed the Food Frequency Questionnaire (FFQ), the Confusion, Hubbub, and Order Scale (CHAOS), and the Behavior Rating Inventory of Executive Function^®^-Preschool Version (BRIEF-P) to assess model variables. Regression analyses revealed a significant and independent association between assorted snacks and processed foods and two EF subscales. There were also significant associations between household chaos and each EF subscale. There was no significant moderation effect. These findings suggest that family households characterized by dysregulation are associated with children’s EF difficulties during early childhood and that the role of unhealthy dietary intake in child EF should be explored further. Future longitudinal studies that include multi-method approaches are needed to document the mechanisms through which household chaos impacts child EF over time.

## 1. Introduction

Executive function (EF) consists of an array of higher-order cognitive skills that have been associated with various forms of optimal functioning [1]. EF abilities involve processes that govern thoughts and behaviors, which can influence how children react to situations and create relationships across contexts. In addition to genetic contributions [2,3,4,5], dietary intake and household environment factors have been shown to influence children’s EF abilities. For example, the overall caregiving environment and household are thought to be critical in promoting the development of children’s EFs [6,7,8,9]. In addition, research has shown that young children who eat more healthy foods and fewer snack foods or processed meats tend to have more optimal EF abilities [10,11]. With respect to household environments, the literature suggests that higher levels of household chaos (settings high in noise and with fewer routines) is associated with lower executive function abilities, such as limited ability to focus or control one’s emotions [12,13], and may indirectly predict behavioral regulation in children [14].

Although these aspects of a child’s environment may influence EF development, there is less known about the interplay between dietary intake and household environment factors as influences on children’s EF. To address this gap in the literature, we tested a model examining parent-reported dietary intake and child EF, and household chaos was tested as a moderator.

### 1.1. Executive Function

EF encompasses general-purpose control mechanisms that are often linked to the brain’s prefrontal cortex (PFC). The literature shows that EF may regulate and influence the dynamics of cognition, actions, and create meaning for various ages [1,15]. These processes develop throughout childhood and may play a role in a child’s social, emotional, and physical interactions and relationships [16,17]. The frontal lobes, specifically in the PFC, send and receive information from major sensory and motor control regions, and the PFC is an essential brain structure that oversees and manages neural systems located in cortical and subcortical areas. The PFC continually monitors activities in the cortical and subcortical regions while sending signals to execute certain behaviors [18]. Thus, these signals can influence an individual’s executive function abilities and behaviors.

The literature presents three core dimensions of EF: Inhibitory Control, Working Memory, and Cognitive Flexibility or set-shifting. These processes can control goal-directed actions and responses to complex or significant situations [19,20]. They may consist of reasoning, problem-solving, and organizing within a child’s life [15,21,22,23]. These EF abilities and skills are crucial for success in school and social groups, cognitive development, and physical and mental health [3]. The ability to alter behavioral responses, shift attention, control emotions or feelings, plan or organize, and use one’s working memory all fall within EF’s dimensions. Inhibitory control includes the ability to control one’s emotions, attention, and behaviors while doing what may be appropriate or needed in a situation [3]. Shift refers to the ability to move from one situation or aspect of a problem to another. Emotional control involves difficulties in emotional expression and a child’s ability to control one’s emotions and feelings. Planning/organizing refers to the child’s ability to manage current or future tasks in various situations. Lastly, working memory involves the capacity to hold information in memory for goals or plans. Collectively, these aspects reflect the broader construct of EF.

### 1.2. Dietary Intake and Children’s EF

Nutrition plays an essential role in a child’s healthy development, including EF capacities. Researchers have found evidence suggesting potential bidirectional links between EF and dietary intake. Riggs et al. (2010) found that in fourth-grade students (M_age_ = 9.4), EF proficiency was negatively related to snack food intake. They discussed how youth with enhanced cognitive abilities and emotional control skills might be better at inhibiting the cognitive and emotional rewards that may come with snack food. They also explain that youth with stronger working memory skills may have more goals to eat healthier foods [24]. This study illustrates that cognition may impact dietary intake on multiple levels. The literature also shows that creative thinking and working memory may be affected by added sugar and dietary fiber. For example, in preadolescent children, added sugar intake was negatively associated with tests of creative thinking, and dietary fiber was positively associated with overall creative thinking [25]. Additionally, researchers found that executive cognition-function in fourth-grade children was negatively associated with high-calorie snack food intake and positively associated with fruit and vegetable intake [26].

A recent systematic review exploring young and old adolescents and older children found a positive association between healthy food intake and EF, including whole grains, fish, fruits, and vegetables. In addition, less nutritious food, including snack foods, sugar-sweetened beverages, and processed meats, were inversely related to EF [10]. The studies measured various dimensions of EF, including inhibition, working memory, and attention and planning. When explicitly examining inhibition in adolescents, researchers found a positive association between inhibitory problems, poor decision-making, and intake of sweet drinks and snack food [27]. In older children and young adolescents, intake of mixed grains was beneficial for cognitive performance [28], and poorer diet quality was associated with worse cognition [29]. In addition, increased fish intake, using a cluster-randomized cross-over trial, explained increases in reading and inattention performance in third- and fourth-grade children (aged 8–11 years) [30].

Overall, study findings suggest positive associations between healthy food intake and children’s EF abilities, and negative associations between snack, processed, or more unhealthy food intake and children’s EF abilities. However, most studies were conducted with older children and adolescent samples and focused on school-related executive function tasks. The current study focuses on a younger population (18–24 months), which can help researchers and community members understand the effect of dietary intake on children’s EFs at a younger age when these skills are rapidly developing. These findings might also shed light on how influential the caregivers can be, primarily when they are making critical food-related decisions for their children.

### 1.3. Household Chaos

Household chaos describes an environment that is high in noise and crowding and low in regularity and routines [14]. Researchers have found that household chaos is associated with various outcomes, such as behavior problems, limited attentional focusing, reduced ability to understand and act in certain social situations, and reduced accuracy and efficiency in cooperative parent-child tasks [12]. Additionally, researchers have found a direct association between higher levels of household chaos and poor performances on tasks that are related to core dimensions of executive functioning, such as inhibitory control, cognitive flexibility, working memory, and effortful control [20,31,32,33,34]. Higher levels of household chaos may also interfere with a child’s proficiency levels, creating a lack of control within their environment [35]. Furthermore, household disorganization may also negatively influence working memory, attention shifting, and other forms of inhibitory control [36], and a lack of routine has often been associated with poor performance on EF tasks in kindergartners [37]. Lastly, a recent meta-analysis found evidence that household chaos is significantly and negatively associated with child executive functioning [31].

Furthermore, positive home environments (low family conflict, high family cohesion, and low household chaos) have been associated with healthier food-related behaviors. More negative home environments (high family conflict, low family cohesion, and increased household chaos) have been associated with more unhealthy food-related behaviors [38,39,40,41,42,43,44,45].

Taken together, prior research has demonstrated that both unhealthy dietary intake in children and chaotic living environments may impact a variety of children’s EF skills. However, there is less examination of how these factors, together, influence EF skills in young children. Dysregulated environments may exacerbate the effects of unhealthy dietary intake on EF, or low household chaos may buffer these effects. Examining the interplay between household chaos and dietary intake and their influence on early EF skills can help shed light on modifiable factors in the environment and diet that can promote developing EF capacities.

### 1.4. Goal of the Present Study

This study aimed to examine the relation between children’s dietary intake, household chaos, and EF abilities. Because these children were 18–24 months of age, we wanted to explore food groups and dietary intake patterns when caregivers are the primary source of food availability. The study also aimed to examine whether household chaos moderates these associations. First, we examined the association between six dietary intake patterns and children’s EFs. We hypothesized that children with a healthier dietary intake would have higher EF scores. Second, we determined whether household chaos moderated the association between dietary intake and EF subscales. We hypothesized that lower household chaos would buffer children from the effects of a less healthy dietary intake.

## 2. Materials and Methods

### 2.1. Participants

The data in this analysis are from families participating in the STRONG Kids2 (SK2) longitudinal birth cohort study in the Midwestern United States. It is designed to examine multi-level predictors of weight trajectories, dietary habits, and family relationships over the first seven years of life [46]. As part of the more extensive study, caregivers completed surveys regarding their household chaos, child EF behaviors, child dietary intake, and demographics for both the caregiver and child. This study was approved by the University Institutional Review Board. Participant characteristics can be found in Table 1.

### 2.2. Measures

#### 2.2.1. Executive Function

EF was assessed using the Behavior Rating Inventory of Executive Function-Preschool Version (BRIEF-P) [17]. This survey assesses multiple dimensions of EF, including inhibition, shifting, emotional control, working memory, and planning/organizing. Parents are asked to complete this questionnaire based on how often each behavior has been an issue in their child’s life during the last six months, and these items are ranged from 1 (never) to 3 (often). Example statements include: “is easily overwhelmed or overstimulated by typical daily activities”, “gets out of control more than playmates”, and “talks or plays too loudly.” Five subscales (listed above), three broad index scores (Inhibitory Self-Control, Flexibility, and Emergent Metacognition), and one global composition score are also obtained from the BRIEF-P. Two validity scales (Inconsistency and Negativity) can also be acquired. The Inhibitory Self-Control Index (ISCI) combines the inhibit and emotional control subscales, the Flexibility Index (FI) is a sum of the shift and emotional control subscales, and the Emergent Metacognition Index (EMI) incorporates the working memory and plan/organize subscales. The global composition score (GEC) consolidates all subscales to receive a total score. Higher scores on the BRIEF-P indicate worse performance on EF abilities. For this study, the ISCI, FI, and EMI indexes will be used for all analyses. Adequate internal reliability was demonstrated for the ISCI (Cronbach’s α = 0.92), FI (Cronbach’s α = 0.88), and EMI subscales (Cronbach’s α = 0.93).

#### 2.2.2. Dietary Intake

Child dietary intake was assessed using a child block of the Food Frequency Questionnaire (FFQ) developed by Nutrition Quest [47]. The FFQ measures the intake of fruits, vegetables, fats, proteins, and dairy in children 2 to 7 years of age. Parents complete the 90-item questionnaire in response to their child’s “usual eating habits in the past six months”. Example items include banana, broccoli, beef, and butter. Questions are rated on an 8-point Likert scale. The scale consists of the following: never, once per month, 2–3 times per month, once per week, twice per week, 3–4 times per week, 5–6 times per week, and every day. The food list was developed from NHANES III dietary recall data, and the USDA Nutrient Database for Standard Reference was also used for nutrient assessment. Based on previous literature, food groups were created to combine related food types [48,49]. This method was used to analyze food groups or profiles instead of specific nutrients specifically. There were 23 food groups, and a principal component analysis was conducted to reduce the dietary intake data. The results revealed six components from the 23 food groups (see description below and Table 2).

#### 2.2.3. Household Chaos

Household chaos and environmental levels were assessed using the Confusion, Hubbub, and Order Scale (CHAOS) [50]. The questionnaire consists of 15 statements surrounding their household environment and chaos. Each question is on a 4-point Likert scale, ranging from “very much like your home” to “not at all like your own home”. Example questions include: “It’s a real zoo in our home”, “Our home is a good place to relax”, and “First thing in the day, we have a regular routine at home”. CHAOS measures two categories of household chaos. The first is routines and organization, and the second facet is disorganization, confusion, and noise. A single score is obtained by summing the items with the highest possible score of 60. A higher score indicates a higher level of chaos within the home or environment. Adequate internal reliability was demonstrated for household chaos (Cronbach’s α = 0.86).

#### 2.2.4. Data Analysis

The Statistical Package for the Social Sciences (SPSS; IBM Corp, Armonk, NY, USA) version 27.0 was used for data analysis, and the statistical significance was set at *p* ≤ 0.05. A PCA was conducted to examine food groups further and create dietary intake model variables. To test associations among all study variables, including demographic variables, bivariate parametric correlations were conducted. Then, a series of multiple regressions were conducted to test the hypothesis that dietary intake and household chaos have a unique and combined impact on executive function. For the moderation analysis, interaction terms were created by multiplying the dietary intake variables and household chaos scores.

## 3. Results

### 3.1. Preliminary Analyses

Preliminary analyses were conducted to determine whether demographic variables were associated with executive function variables. Child gender, family income, and parent race/ethnicity were not associated with any EF outcome variables (all *p*’s > 0.05). Since there were no significant correlations, no demographic variables were used in further analyses.

### 3.2. FFQ Analyses

To create dietary intake food groups, a principal components analysis (PCA) was conducted. A PCA with varimax rotation was performed using responses to the food groups formed from the FFQ. Six components with Eigenvalues > 1.00 collectively accounted for 57.365% of the variance. A factor loading of 0.4 was used, which was consistent with previous studies [51,52,53,54]. All food groups loaded on a factor using this cutoff. Components can be found in Table 2.

Component 1, Assorted Snacks and Processed Foods, explained 13.93% of the variance and included Savory Snacks, Refined Carbohydrates, Fried Foods, Processed Meats, Mixed Foods, Condiments, and Butter/Margarine. Component 2, Assorted Vegetables, Fruit, and Fish, explained 11.53% of the variance and included Vegetables, Starchy Foods, Legumes, Fruit, and Fish/Seafood. Component 3, Fruit Juice and Sweet Items, contained 100% Fruit Juice, Sweet Beverages, and Sweet Foods. This explained 8.98% of the variance. Component 4, Assorted Proteins, included Poultry, Red Meats, and Eggs, and it explained 8.76% of the variance. Component 5, Grains and Nuts, explained 7.80% of the variance, and contained items directly related to Grains and Peanuts/Nuts. Lastly, Component 6, Assorted Dairy and Water, included Yogurt, Dairy, and Water and explained 6.37% of the variance. Adequate internal reliability was demonstrated for Component 1 (Cronbach’s α = 0.76), Component 2 (Cronbach’s α = 0.65), and Component 3 (Cronbach’s α = 0.55). We used a cutoff of 0.55 due to our sample size and to have adequate reliability for each component [55,56,57]. Therefore, Component 4 (Cronbach’s α = 0.51), Component 5 (Cronbach’s α = 0.47), and Component 6 (Cronbach’s α = 0.26) were not used in the final analyses.

### 3.3. Bivariate Correlations

#### Dietary Intake and Child EF

Bivariate associations between all model variables are depicted in Table 3. Assorted Snacks and Processed Foods were positively and significantly associated with household chaos and each EF subscale (all *p*’s < 0.05). Intake of Assorted Vegetables, Fruit, and Fish was negatively and significantly associated with household chaos r (331) = −0.12, *p* = 0.03, and no EF subscales. Fruit Juice and Sweet Items were positively and significantly associated with household chaos and all EF subscales (all *p*’s < 0.05). Assorted Snacks and Processed Foods, and Fruit Juice and Sweet Items were the only subscales used in the final analyses.

### 3.4. Regression Analysis

#### Dietary Intake and Child EF

In the second analysis, dietary intake, household chaos, and the interaction term were entered as predictor variables, and each EF index was tested separately as the dependent variable. Regarding the Assorted Snacks and Processed Foods subscale, results revealed that the whole model accounted for about 20% of the variance (Adjusted *R*^2^ = 0.20). For the EF ISCI index, household chaos (β = 0.41, *p* ≤ 0.000) and the Assorted Snacks and Processed Foods subscale (β = 0.13, *p* = 0.02) were both significant predictors; however, the interaction variable was not significant (see Table 4). Similar results were found for the EMI index (Table 5). However, for the EF FI index, the Assorted Snacks and Processed Foods subscale was not a significant predictor in the model (Table 6). Regarding the Fruit Juice and Sweet Items subscale, results revealed that the whole model accounted for about 18% of the variance (Adjusted *R*^2^ = 0.18). For the EF ISCI index, only household chaos (β = 0.41, *p* ≤ 0.000) was a significant predictor of EF abilities (see Table 7). Similar results were found for EMI and FI indexes (Table 8 and Table 9).

## 4. Discussion

This study examined associations between dietary intake, household chaos, and EF in children 18–24 months of age. In the final analyses, a significant association between dietary intake and EF for the Assorted Snacks and Processed Foods component was obtained, with higher intake related to lower shift and emotional control abilities, as well as lower working memory and planning and organizing abilities. No other independent associations with other dietary intake variables were observed. Contrary to our hypotheses, results also indicated that household chaos does not modify dietary intake and EF associations but instead has an independent effect on EF. Additionally, household chaos was significantly associated with all three BRIEF-P indexes in the final models. The association between household chaos and poorer EF is consistent with previous literature [12,58,59]. Because household chaos describes an environment that is high in noise and crowding and low in regularity and routines [14], this environment may distract, limit, or alter a child’s EF abilities.

The current findings can contribute to the larger literature in multiple ways. First, we found independent associations between household chaos and child EF, and this highlights how even at a young age (18–24 months) household chaos might impact children’s EF abilities. They may not understand the signals around them when environments are noisy or disorganized, and the lack of routine and regularity might influence their attentional and emotional regulation. Second, these findings can help researchers, policymakers, and families understand how household chaos might influence children’s dietary intake and EF abilities. Informing other scholars, individuals, and policymakers about this phenomenon can help families develop more routines and healthier lifestyles and provide knowledge about how external influences can impact children’s executive functions. Lastly, these results can inform future studies that set out to examine how household chaos may affect other dimensions of EF using various methodologies, designs, and more diverse samples.

With respect to dietary intake, our hypotheses were partially correct regarding associations between Assorted Snacks and Processed Foods and worse performance in all EF subscales. We also saw a similar pattern of relations at a correlational level between the Fruit Juice and Sweet Items subscale and all EF subscales. This indicates that various types of food may impact a child’s performance in EF-related tasks. Researchers found, using NHANES data, that nearly all children aged 2 to 5 years old consumed a snack on a specific day, with 62% occurring in the morning, 84% in the afternoon, and 72% in the evening. They found that these snacks accounted for 28% of total energy intake, and many of these snacks also included beverages consumed within a specific day [60]. These findings further illustrate that snacks and certain beverages are prominent within young children and how we need to be more aware of how influential snacks and beverages can be to a young child’s diet and lifestyle. Consistent with previous research [10,24,26,27,29], diet quality may influence a child’s or adolescent’s performance in EF-related tasks, and this may further influence their cognitive abilities. This warrants further investigation to understand how diet quality affects younger children’s developing EF capacities. Many caregivers assist with feeding when children are young, so future researchers need to examine further how this relationship and assistance might influence a child’s dietary intake.

There are also limitations to this study. The sample was not diverse with respect to race/ethnicity and income. Therefore, the findings are not generalizable, and diverse family contexts and relationships should be examined in the future. No causal inferences can be made regarding the discovered associations based on the correlational design, highlighting the need for longitudinal and experimental methodologies. Future research should incorporate multiple measures of dietary intake, EF, and household chaos to understand these results further.

Despite these limitations, the current findings are novel and contribute to the literature on EF in children aged 18 to 24 months. For example, they further support the notion that the nature of the household environment, specifically household chaos, may influence young children’s developing EF capacities. As such, preventions focused on activities and support for parents to establish healthy routines and lower unhealthy food intake in their children might help mitigate EF problems. Furthermore, the data suggest that children’s unhealthy dietary intake may also be related to the household environment, and these associations warrant future study. These factors might impact a child and family on multiple levels and have implications for children’s developing EF capacities.

## Figures and Tables

**Table 1 nutrients-13-04442-t001:** Descriptive Statistics for Model Variables.

	N	%	M	SD
Child gender				
Male	142	49.0		
Female	148	51.0		
Caregiver monthly income				
$3000 and under	53	18.3		
$3001–$5000	77	26.6		
$5001 and above	136	46.9		
Prefer not to say	24	8.3		
Perceived income hardship (at the end of the month)				
More than enough money left	148	69.2		
Some money left	45	21.0		
Just enough money left	12	5.6		
Somewhat short of money	8	3.7		
Very short of money	1	0.5		
Caregiver work schedule				
Full-time	154	72.0		
Part-time	60	28.0		
Average work hours	214		34.52	12.05
Caregiver marital status				
Single	12	5.6		
Civil union	1	0.5		
Married	194	90.7		
Co-habituating	6	2.8		
Divorced	1	0.5		
Caregiver age	225		31.33	4.33
Caregiver race/ethnicity				
American Indian/Alaska Native	2	0.7		
Asian	21	7.4		
Biracial	7	2.5		
Black	15	5.3		
White	234	83.0		
Prefer not to say	3	1.1		
Child medical condition				
Yes	6	2.8		
No	208	97.2		
Household chaos	294		26.69	7.25
ISCI score	275		38.50	8.13
FI score	294		29.37	5.82
EMI score	270		38.70	8.59
Assorted snacks and processed foods	342		11.92	2.65
Assorted vegetables, fruit, and fish	342		14.50	3.38
Fruit juice and sweet items	342		9.16	3.23
Assorted proteins	342		5.29	1.81
Grains and nuts	342		8.86	2.77
Assorted dairy and water	342		10.65	1.80

Note. The Inhibitory Self-Control Index (ISCI) includes inhibit and emotional control subscales. The Flexibility Index (FI) includes shift and emotional control subscales. The Emergent Metacognition Index (EMI) includes working memory and plan/organize subscales.

**Table 2 nutrients-13-04442-t002:** Dietary Intake: Principal Component Analysis.

Description	Loading
**Assorted Snacks and Processed Foods**	
Savory snacks	0.69
Refined carbohydrates	0.69
Fried foods	0.62
Processed meats	0.54
Mixed foods	0.54
Condiments	0.51
Butter/Margarine	0.49
**Assorted Vegetables, Fruit, and Fish**	
Vegetables	0.81
Starchy foods	0.68
Legumes	0.65
Fruit	0.60
Fish/Seafood	0.50
**Fruit Juice and Sweet Items**	
100% Fruit juice	0.76
Sweet beverages	0.65
Sweet foods	0.55
**Assorted Proteins**	
Poultry	0.68
Red meats	0.67
Eggs	0.57
**Grains and Nuts**	
Grains	0.69
Peanuts/Nuts	0.67
**Assorted Dairy and Water**	
Yogurt	0.80
Dairy	0.59
Water	0.47

Note. A factor loading cutoff of 0.4 was used for analyses.

**Table 3 nutrients-13-04442-t003:** Pairwise Bivariate Correlations Among all Model Variables.

		1	2	3	4	5	6	7
1	Assorted snacks and processed foods	----						
2	Assorted vegetables, fruit, and fish	**0.32 *****	----					
3	Fruit juice and sweet items	**0.56 *****	0.09	----				
4	Household chaos	**0.17 ****	**−0.12 ***	**0.23 *****	----			
5	ISCI subscale	**0.20 *****	−0.10	**0.17 ****	**0.38 *****	----		
6	FI subscale	**0.16 ****	−0.04	**0.13 ***	**0.25 *****	**0.86 *****	----	
7	EMI subscale	**0.19 ****	−0.05	**0.13 ***	**0.38 *****	**0.82 *****	**0.67 *****	----

Note. Statistically significant correlations are bolded (*p* ≤ 0.05 *, *p* ≤ 0.01 **, *p* ≤ 0.001 ***).

**Table 4 nutrients-13-04442-t004:** Multiple Regression Analyzing Associations Between Assorted Snacks and Processed Foods, Household Chaos, and Inhibitory Self-Control Index (ISCI).

	Model 1		Model 2		Model 3	
	B (SE)	β (*p*)	B (SE)	β (*p*)	B (SE)	β (*p*)
Assorted snacks and processed foods	0.64 (0.19)	**0.20 (0.001)**	0.40 (0.18)	**0.12 (0.03)**	0.40 (0.18)	**0.13 (0.24)**
Household chaos			0.46 (0.06)	**0.41 (0.000)**	0.46 (0.06)	**0.41 (0.000)**
Assorted snacks and processed foods × household chaos					0.03 (0.02)	0.07 (0.20)
*R* ^2^	0.04		0.20		0.20	
Δ*R*^2^	0.04		0.19		0.20	
ΔF	**11.07**		**54.28**		1.65	

Note. Bolded values indicate statistically significant findings (*p* ≤ 0.05); *N* = 275; Assorted snacks and processed foods component includes Savory Snacks, Refined Carbohydrates, Fried Foods, Processed Meats, Mixed Foods, Condiments, and Butter/Margarine.

**Table 5 nutrients-13-04442-t005:** Multiple Regression Analyzing Associations Between Assorted Snacks and Processed Foods, Household Chaos, and Emergent Metacognition Index (EMI).

	Model 1		Model 2		Model 3	
	B (SE)	β (*p*)	B (SE)	β (*p*)	B (SE)	β (*p*)
Assorted snacks and processed foods	0.65 (0.21)	**0.19 (0.002)**	0.43 (0.20)	**0.13 (0.03)**	0.43 (0.20)	**0.13 (0.03)**
Household chaos			0.45 (0.07)	**0.37 (0.000)**	0.45 (0.07)	**0.37 (0.000)**
Assorted snacks and processed foods × household chaos					0.02 (0.03)	0.04 (0.52)
*R* ^2^	0.04		0.17		0.17	
Δ*R*^2^	0.03		0.16		0.16	
ΔF	**9.74**		**42.96**		0.43	

Note. Bolded values indicate statistically significant findings (*p* ≤ 0.05); *N* = 270; Assorted snacks and processed foods component includes Savory Snacks, Refined Carbohydrates, Fried Foods, Processed Meats, Mixed Foods, Condiments, and Butter/Margarine.

**Table 6 nutrients-13-04442-t006:** Multiple Regression Analyzing Associations Between Assorted Snacks and Processed Foods, Household Chaos, and Flexibility Index (FI).

	Model 1		Model 2		Model 3	
	B (SE)	β (*p*)	B (SE)	β (*p*)	B (SE)	β (*p*)
Assorted snacks and processed foods	0.36 (0.13)	**0.16 (0.01)**	0.24 (0.13)	0.11 (0.06)	0.24 (0.13)	0.11 (0.06)
Household chaos			0.23 (0.05)	**0.28 (0.000)**	0.23 (0.05)	**0.29 (0.000)**
Assorted snacks and processed foods × household chaos					0.02 (0.02)	0.07 (0.25)
*R* ^2^	0.02		0.10		0.11	
Δ*R*^2^	0.02		0.10		0.10	
ΔF	**7.24**		**25.00**		1.35	

Note. Bolded values indicate statistically significant findings (*p* ≤ 0.05); *N* = 294; Assorted snacks and processed foods component includes Savory Snacks, Refined Carbohydrates, Fried Foods, Processed Meats, Mixed Foods, Condiments, and Butter/Margarine.

**Table 7 nutrients-13-04442-t007:** Multiple Regression Analyzing Associations Between Fruit Juice and Sweet Items, Household Chaos, and Inhibitory Self-Control Index (ISCI).

	Model 1		Model 2		Model 3	
	B (SE)	β (*p*)	B (SE)	β (*p*)	B (SE)	β (*p*)
Fruit juice and sweet items	0.43 (0.16)	**0.17 (0.01)**	0.18 (0.15)	0.07 (0.22)	0.19 (0.15)	0.07 (0.21)
Household chaos			0.47 (0.06)	**0.41 (0.000)**	0.47 (0.06)	**0.41 (0.000)**
Fruit juice and sweet items × household chaos					0.01 (0.02)	0.02 (0.72)
*R* ^2^	0.03		0.19		0.19	
Δ*R*^2^	0.02		0.18		0.18	
ΔF	**7.63**		**54.06**		0.13	

Note. Bolded values indicate statistically significant findings (*p* ≤ 0.05); *N* = 275; Fruit juice and sweet items component contains 100% Fruit Juice, Sweet Beverages, and Sweet Foods.

**Table 8 nutrients-13-04442-t008:** Multiple Regression Analyzing Associations Between Fruit Juice and Sweet Items, Household Chaos, and Emergent Metacognition Index (EMI).

	Model 1		Model 2		Model 3	
	B (SE)	β (*p*)	B (SE)	β (*p*)	B (SE)	β (*p*)
Fruit juice and sweet items	0.34 (0.16)	**0.13 (0.04)**	0.10 (0.15)	0.04 (0.53)	0.10 (0.16)	0.04 (0.52)
Household chaos			0.46 (0.07)	**0.38 (0.000)**	0.46 (0.07)	**0.38 (0.000)**
Fruit juice and sweet items × household chaos					0.004 (0.02)	0.01 (0.87)
*R* ^2^	0.02		0.16		0.16	
Δ*R*^2^	0.01		0.15		0.15	
ΔF	**4.50**		**43.78**		0.03	

Note. Bolded values indicate statistically significant findings (*p* ≤ 0.05); *N* = 270; Fruit juice and sweet items component contains 100% Fruit Juice, Sweet Beverages, and Sweet Foods.

**Table 9 nutrients-13-04442-t009:** Multiple Regression Analyzing Associations Between Fruit Juice and Sweet Items, Household Chaos, and Flexibility Index (FI).

	Model 1		Model 2		Model 3	
	B (SE)	β (*p*)	B (SE)	β (*p*)	B (SE)	β (*p*)
Fruit juice and sweet items	0.23 (0.11)	**0.13 (0.03)**	0.11 (0.11)	0.06 (0.30)	0.12 (0.11)	0.07 (0.26)
Household chaos			0.23 (0.05)	**0.29 (0.000**)	0.23 (0.05)	**0.28 (0.000)**
Fruit juice and sweet items × household chaos					0.02 (0.02)	0.07 (0.24)
*R* ^2^	0.02		0.09		0.10	
Δ*R*^2^	0.01		0.09		0.09	
ΔF	**4.77**		**24.98**		1.40	

Note. Bolded values indicate statistically significant findings (*p* ≤ 0.05); *N* = 294; Fruit juice and sweet items component contains 100% Fruit Juice, Sweet Beverages, and Sweet Foods.

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
