# Peer review of "The Impact of Household Chaos and Dietary Intake on Executive Function in Young Children"

_nutrients, 2021, doi:10.3390/nu13124442_

Round 1

Reviewer 1 Report

The content of this study is interesting. I have some considerations to improve your work. In the abstract you explain that the sample is of 275 families . In the part of the participants, 402 children are described. Are some of them brothers? Please clarify this point. It would be useful to have more information about the participants (neurodevelopmental profile, school placement, language, social class, age and profession of parents).

Reviewer 2 Report

This paper raises the interesting theme of analyzing the effects of diet and living environment on executive function. However, the hypothesis is not well established, and the content of the dissertation is poor. The author hypothesized and tested whether the living environment moderates the relationship between diet and executive function. However, the content written on lines 136-147 is not related to moderation, but to the relationship between living environment and food. In other words, the grounds for moderation have not been thoroughly discussed. Furthermore, as a result, no moderation effect was seen, so it makes me wonder if it was necessary to put moderation in the hypothesis in the first place. If the hypothesis is weak and the results are weak, it seems to be of low value as a dissertation. Therefore, I recommend hypothesizing and verifying mediation rather than moderation. Table 3 shows that diet and living environment correlates. Looking at Tables 4 to 9, in some models, both diet and living environment correlates with executive function, while in other models, only living environment correlates with executive function. The former indicates partial mediation and the latter indicates full mediation. The multiple regression model should be as follows. First, as Step 1, put the diet in the independent variable and the living environment in the dependent variable. In Step 2, the diet is set as the independent variable and the executive function is set as the dependent variable. Next, put the living environment in the independent variable and the executive function in the dependent variable. In Step 3, the diet and living environment are set as independent variables, and the living environment is set as the dependent variable. In this way, the diet affects the living environment (Step 1), both the diet and the living environment affect the executive function (Step 2), and the living environment completely or partially relates the relationship between the diet and the executive function. Mediation (Step 3) is shown. Studies showing mediation effects using multiple regression analysis in this way include, for example:

Kokubun K, Nemoto K and Yamakawa Y (2020) Fish Intake May Affect Brain Structure and Improve Cognitive Ability in Healthy People. Front. Aging Neurosci. 12:76. doi: 10.3389/fnagi.2020.00076

Table 2 shows that "Assorted vegetables, fruit, and fish" correlate with the living environment. This indicates that the living environment fully mediates the relationship between "Assorted vegetables, fruit, and fish" and cognitive function. "Assorted vegetables, fruit, and fish" should be added to the multiple regression analysis.

Lines 251-253 state that the two factors are not used, but do not state the rationale. If 0.5 is the standard, the rationale should be clearly shown. Or, if 0.55 is used as the standard, the fourth factor can be omitted and the discussion afterward will be simplified. The literature showing 0.55 as a reference is as follows. 

Taber, K.S. The Use of Cronbach’s Alpha When Developing and Reporting Research Instruments in Science Education. Res Sci Educ 48, 1273–1296 (2018). https://doi.org/10.1007/s11165-016-9602-2

Nehring, A., Nowak, K. H., zu Belzen, A. U., & Tiemann, R. (2015). Predicting students’ skills in the context of scientific inquiry with cognitive, motivational, and sociodemographic variables. International Journal of Science Education37(9), 1343-1363.

Round 2

Reviewer 2 Report

My opinion that mediation is better was rejected, but I agree with acceptance because the authors explained the significance of testing moderation. 

Author Response

Thank you again for providing that suggestion and feedback. We greatly appreciated the reviewer’s insight.